# Levodopa-Carbidopa Intestinal Gel in Advanced Parkinson’s Disease: Observations and Dilemmas after 10 Years of Real-Life Experience

**DOI:** 10.3390/pharmaceutics14061115

**Published:** 2022-05-24

**Authors:** József Attila Szász, Viorelia Adelina Constantin, Károly Orbán-Kis, Ligia Ariana Bancu, Simona Maria Bataga, Marius Ciorba, Előd Nagy, Mircea Radu Neagoe, István Mihály, Róbert Máté Szász, Krisztina Kelemen, Mihaela Simu, Szabolcs Szatmári

**Affiliations:** 12nd Clinic of Neurology, Târgu Mureș County Emergency Clinical Hospital, 540136 Târgu Mureș, Romania; szaszneuro@yahoo.com (J.A.S.); vioreliaconstantin@yahoo.com (V.A.C.); krisztinn.kelemen@gmail.com (K.K.); szabolcs.szatmari@umfst.ro (S.S.); 2Department of Neurology, George Emil Palade University of Medicine, Pharmacy, Science and Technology of Târgu Mureș, 540139 Târgu Mureș, Romania; szaszrobert48@gmail.com; 3Doctoral School, ”Victor Babes” University of Medicine and Pharmacy Timisoara, 300041 Timisoara, Romania; 4Department of Physiology, George Emil Palade University of Medicine, Pharmacy, Science and Technology of Târgu Mureș, 540139 Târgu Mureș, Romania; istvan.mihaly@umfst.ro; 5Department of Internal Medicine, George Emil Palade University of Medicine, Pharmacy, Science and Technology of Târgu Mureș, 540139 Târgu Mureș, Romania; ligiabancu@yahoo.com; 61st Clinic of Internal Medicine, Târgu Mureș County Emergency Clinical Hospital, 540142 Târgu Mureș, Romania; 7Department of Gastroenterology, George Emil Palade University of Medicine, Pharmacy, Science and Technology of Târgu Mureș, 540139 Târgu Mureș, Romania; simona.bataga@umfst.ro (S.M.B.); ciorba_im@yahoo.com (M.C.); 8Department of Gastroenterology, Târgu Mureș County Emergency Clinical Hospital, 540142 Târgu Mureș, Romania; 9Department of Biochemistry, George Emil Palade University of Medicine, Pharmacy, Science and Technology of Târgu Mureș, 540139 Târgu Mureș, Romania; elod.nagy@umfst.ro; 10Laboratory of Medical Analysis, Clinical County Hospital Mures, 540142 Târgu Mureș, Romania; 11Department of Surgery, George Emil Palade University of Medicine, Pharmacy, Science and Technology of Târgu Mureș, 540139 Târgu Mureș, Romania; radu.neagoe@umfst.ro; 122nd Clinic of Surgery, Târgu Mures County Emergency Clinical Hospital, 540136 Târgu Mures, Romania; 13Department of Neurology, Emergency County Hospital Miercurea Ciuc, 530173 Miercurea Ciuc, Romania; 14Department of Neurology, ”Victor Babes” University of Medicine and Pharmacy Timisoara, 300041 Timisoara, Romania; simu.mihaela@umft.ro; 15”Pius Branzeu” Emergency Clinical County Hospital, 300723 Timisoara, Romania

**Keywords:** advanced Parkinson’s disease, levodopa/carbidopa intestinal gel, device aided therapy, motor fluctuations, dyskinesia, dopamine agonist, estimated dose

## Abstract

Advanced Parkinson’s disease (APD) cannot be treated efficiently using the classical medications however, in recent decades invasive therapeutical methods were implemented and confirmed as effective. One of these methods makes it possible to continue the levodopa (LD) supplementation as a gel administered directly into the upper intestine. However, there are a number of unanswered questions regarding this method. Therefore, we retrospectively analyzed a 10-year period of selected patients that were treated with levodopa/carbidopa intestinal gel (LCIG). We included all APD patients with motor fluctuations and dyskinesia at presentation. LCIG treatment was started in 150 patients: on average these patients received LD for 10.6 ± 4.4 years with a frequency of 5.2 ± 1.0/day until the introduction of LCIG. The estimated and the real LCIG dose differed significantly (mean: 1309 ± 321 mg vs. 1877 ± 769 mg). The mean duration of LCIG administration was 19.8 ± 3.6 h, but in a number of 62 patients we had to administer it for 24 h, to maximize the therapeutic benefit. A carefully and individually adjusted LCIG treatment improves the quality of life of APD patients, but questions remain unresolved even after treating a large number of patients. It is important to share the ideas and observations based on the real-life experience related to the optimal timing, the appropriate dose and duration of administration of the LCIG.

## 1. Introduction

Parkinson’s disease (PD) is a neurodegenerative disease with significantly improved therapy in recent decades; therefore even patients with advanced PD (APD) now have access to effective treatment options, even though these represent invasive methods [1,2]. In the advanced stages of the disease, the efficacy of oral levodopa (LD) treatment (usually combined with an aromatic amino acid decarboxylase inhibitor) is gradually decreased and becomes unpredictable. After oral administration, LD is converted to dopamine within the dopaminergic neurons and stored and released close to a physiological manner. As the neurons degenerate, the administered LD is converted by the remaining dopaminergic terminals, but also by other cells (mainly serotoninergic) that have little or no capacity to store dopamine. The ability to buffer levodopa plasma concentrations is progressively lost, causing a pulsatile stimulation of the postsynaptic dopaminergic receptors. Therefore, the orally administered intermittent doses induce discontinuous stimulation of these receptors, which is hypothesized to lead to molecular changes (aberrant plasticity) finally leading to the development of disabling motor fluctuations and dyskinesias in most patients [3,4]. In this context, one can assume that a continuous delivery of levodopa may be able to alleviate motor complications.

The levodopa/carbidopa intestinal gel (LCIG), containing levodopa (20 mg/mL) and carbidopa monohydrate (5 mg/mL) is a treatment developed for patients with advanced PD (APD). The LCIG delivered continuously into the upper intestine via percutaneous endoscopic gastro-jejunostomy (PEG-J) provides more stable plasma levels compared to oral LD [5,6,7]. Timely introduction of LCIG therapy may improve the quality of life (QoL) of patients with symptoms poorly controlled with maximally optimized oral/transdermal therapy. Currently there are no universally standardized and validated criteria to define APD [8] or to identify patients suitable for LCIG therapy (i.e., predictable efficiency and long-term tolerability with added maximum safety). This is the reason why there is a lack of real-life studies regarding the characteristics of APD patients that can benefit most from this therapeutic option. More precisely we lack data regarding the limitations of the last conventional dopaminergic treatments (dosage of LD, add-on treatments, etc.) as well as regarding the severity and subtypes of motor complications [9,10]. In this paper, we aim to analyze the characteristics/limitations of the latest conservative treatments and review the challenges of initiating LCIG therapy (eligibility assessment, titration process, and continuous dose adjustment for maximum benefit).

## 2. Materials and Methods

We retrospectively evaluated all APD patients with motor fluctuations and dyskinesia (peak dose, diphasic dyskinesia, end of dose dystonia) at presentation, that were treated between June 2011 and June 2021 with LCIG by the multidisciplinary team of the 2nd Neurology Clinic in Târgu Mureș. In this tertiary center the only available device-aided therapy during the evaluated period was LCIG. We recorded the demographic and clinical aspects, disease and treatment duration, severity (both durin *on* and *off* phase) as measured on the Hoehn and Yahr scale, and on the MMSE (Mini Mental State Evaluation) score. The exact doses of LD, dopamine agonists, monoamine oxidase B inhibitor (MAO-Bi), catechol-O-methyl transferase inhibitor (COMTi) and/or amantadine and any kind of motor fluctuations and dyskinesias were recorded as well. For testing the efficiency of LCIG therapy and also for the adjustment of titration of doses patients were hospitalized. Calculation of estimated LCIG doses (LCIG calculated/theoretic) was performed according to literature recommendations [11]. During the titration period and also after performing the PEG-J, we continuously adjusted the doses, in order to maximize the therapeutic benefit. At discharge all patients completed a Patient Global Impression of Improvement (PGI-I) scale to rate their total subjective improvement of PD, regardless if it is due entirely to drug treatment. The improvement was graded on a seven-step scale: 1. Very much improved, 2. Much improved, 3. Minimally improved, 4. No change, 5. Minimally worse, 6. Much worse, 7. Very much worse.

Statistical analysis was performed using the Prism 8.0 software package (GraphPad Software, San Diego, CA, USA). Depending on the type of data, descriptive statistics, Fisher’s exact and Mann—Whitney tests were used, the level of statistical significance was *p* < 0.05. The study was approved by the local ethical committee (UMFST 94/19.05.2017).

## 3. Results

During the aforementioned 10 years period we started LCIG treatment in a total of 150 patients out of which 81 (54%) were male. The mean age of the patients was 63.97 ± 8.16 years, and the MMSE score was 26.1 ± 2.39 points. Between the onset of the disease and the introduction of LCIG 10.95 ± 4.44 years elapsed, with half of the patients deciding to accept therapy in less than one month. The spectrum of motor complications (fluctuations and different subtypes of dyskinesia) is presented in Table 1.

The last utilized add-on therapy is detailed in Table 2. Dopamine agonist utilisation in our group of patients was in the upper zone of the effective therapeutic range.

Before the introduction of LCIG patients received LD treatment for a period of 10.6 ± 4.4 years until; in the same period levodopa was divided in 5.2 ± 1.0 doses per day. The last LD dose before PEG-J, the calculated and final LCIG dose which offer a significant improvement of clinical status, both during the *on* and *off* states, according to the Hoehn and Yahr scale (*p* < 0.001) are all presented in Table 3.

The initial parameters and dosing of LCIG therapy (total daily dose and dosing time) are shown in Table 4.

The difference between the theoretically calculated LCIG and the actual dose used (which was necessary to achieve adequate clinical improvement) is presented in Figure 1.

## 4. Discussion

Parkinson’s disease is the most common movement disorder, besides mitochondrial dysfunction and age-related synuclein aggregation the role of neuroinflammation and increased oxidative stress can also be suspected [12,13]. In Parkinson’s disease, uniquely among the wide range of neurodegenerative diseases, a large number of long-term effective symptomatic therapies are available. Symptomatic PD treatment targets the dopaminergic deficit caused by the extensive degeneration of dopaminergic neurons in the substantia nigra pars compacta. The use of levodopa as dopamine-replacement therapy is highly effective and remains the standard drug with which other therapies are compared [14]. However, as the disease progresses, there may be a need for a more personalized approach and fine-tuning, in accordance with the patients’ needs and due to the inevitable onset of disabling motor complications (after 5 years of treatment in about 40% of patients, after 10 years in more than 90% of patients) [4].

Improving the adverse pharmacokinetics of levodopa is a constant challenge [15,16,17], with several alternatives widely used. Further positive effects are expected from the introduction of the new peripheral catechol-O-methyltransferase inhibitor (COMT-i) opicapone [18] and the reversible monoamine oxidase B inhibitor (MAO-Bi) and glutamate modulator safinamide [19,20]. The latter are not yet available in Central and Eastern Europe, including Romania [21,22]. Levodopa bioavailability and its maximum concentration is higher following opicapone administration (provided higher LD plasma levels compared with entacapone). In a recently published study opicapone was not only effective in motor fluctuations improvement, but also safe and well-tolerated at the same time improving global non-motor symptom burden in PD patients (mainly sleep, fatigue, mood, gastrointestinal symptoms and pain) [23]. Dopaminergic tone improvement is further enhanced by various transdermal dopaminergic tone enhancers (either in current clinical practice or in clinical study phase) [24], which improve the clinical picture by avoiding the common gastroparesis in APD patients [25]. Impaired gastrointestinal motility is also “shorted” by the administration of LD through an inhaler, which can quickly and effectively improve the wearing off state [26].

As is often observed in a hospital setting, different add-on agents have different degrees and qualities of beneficial (or detrimental) effects on each patient. The extent to which the newer preparations improve the clinical picture in the late stages (where they are also worth trying) and the extent to which they are suitable for delaying the introduction of various device aided therapies is not yet clear [9].

There are few data in the literature on the incidence and severity of various motor and non-motor complications in patients with APD and the proportion of patients considered to be eligible for device-aided therapies in movement disorders centers [9,27]. In the OBSERVE-PD study (a cross-sectional, observational, multicenter study carried out in 128 movement disorder centers from 18 countries involving 2615 patients) published in 2019, the percentage of patients with APD was 51.3% [28], which varied regionally and ranged from 24% to 82%. This broad range could reflect differences between countries or study populations [29,30].

Similarly, only a small number of publications aim to evaluate the real-life data of patients with APD (the proportion of patients considered eligible for LCIG in tertiary centers serving a large number of patients). Under these conditions, the clinician who provides long-term monitoring of patients with advanced disease and is forced to constantly adjust the therapeutic strategy may face the several clinical practice challenges:What are the upper limits of conventional dopaminergic medication, especially LD doses (maximum daily dose or dosage intervals), without compromising patient/relative compliance, or how are these influenced by access to various add-on therapies or device-aided therapies?How long is it advisable to “insist” on available combination therapy options and what doses should be used in case of these treatments (the concept of “optimized medication”); also, in what way are these treatment options influenced by access (or lack of) to different device-aided therapy options?How can the severity and profile of various motor and non-motor complications be accurately assessed and what is their real impact on quality of life?How can one recognize the time when motor complications can no longer be controlled or ameliorated with standard oral/transdermal therapy (the concept of “medically refractory motor complications”)?What is the best time to start LCIG so that the benefits are maximized (predictability of clinical response and long-term safety)?

Although LCIG therapy was registered in Romania in the fall of 2009 [7], we used this method for our first patient in June 2011. In our previous publications, we presented in detail the selection criteria used by our multidisciplinary team, the specifics and limitations of the use of substitution therapy and available add-on dopaminergic alternatives [9,31]. However, more than four years later, at the end of 2015, the first expert recommendation was issued to standardize APD management. Accordingly, if a patient has a history of 2 h *off* and/or 1 h of severe dyskinesia despite 5 times daily combination levodopa therapy, one of the available device-aided therapies should be considered [32]. In the first six years of the use of LCIG in our clinic, this particular therapy was started in case of a more serious clinical presentation [9]. This trend is also reflected in the ten-year data: an average of 4.73 ± 1.08 h of *off* period in 150 patients and 2.96 ± 0.84 h of dyskinesia in 95 cases (out of which 44 patients presented an average of 3 ± 0.82 h of diphasic dyskinesia) can be found. This system of criteria, which became well-known as the “5-2-1 rule”, provided a good basis for general neurologists working in non-specific centers. However, due to the lack of clear-cut protocols, the “5-2-1 rule” was supplemented by later resolutions [33,34]. Our current analysis (in press [35]) shows that these clarifications are already reflected in everyday practice, with LCIG therapy being introduced sooner in APD patients.

In our previous publications we also showed that available add-on therapies were used significantly more frequently in our patients immediately prior to the introduction of LCIG [9]. This trend is also seen when processing ten years of data from our clinic. In Figure 2 we compare our own data with other available literature databases: the GLORIA Registry (data from 375 patients treated with LCIG in 18 countries) [5], the OBSERVE-PD study (mentioned earlier) and the COSMOS study (real-world, multinational observational study investigating comedication use with LCIG) [36]. Presumably, over the years, various deficiencies in therapy (the lack of some add-on alternatives and/or device-aided therapy types) have led to the development of therapeutic “habits” that are difficult to overcome, maintained both by patient and cultural perception and the social acceptance of each of the device-aided therapy options. In our opinion it is difficult to estimate the effect of the more intensive use of the available add-on medication on the future introduction of LCIG (predictability and long-term safety). The increased use of dopamine agonists is particularly important in this context. When insisting on the use of dopamine agonists, one should note that besides the well-known side effects (impulse control disorders, sudden sleep onset, hallucinations, psychosis) they may increase the incidence of falls; also, the incidence of psychiatric side effects generally increases with age [37].

Previous studies have shown that nearly two-thirds of patients with PD have taken less than 80% of the medications prescribed by their physician [38]. Further deterioration of therapeutic fidelity is expected during disease progression. In addition to the basic LD therapy (which can be administered 5–8 times a day or even more frequently in APD) and a significant number of additional therapies (dopamine agonists, MAO-Bi, COMT-i), therapy for the non-motor complications and other non-related pathologies can cause drug interactions that may further aggravate or complicate the situation. In addition to these numerous challenges, the treating physician should continuously assess the predominantly non-motor symptoms (cognitive decline, depression, apathy, anhedonia) that are expected to further reduce therapeutic adherence, and which can be aggravated by the lack of a close family member and/or institutional care [10,39].

When discussing the optimal use of LCIG, everybody should be aware that there is a lack of therapeutic recommendations to select the appropriate device-aided therapy. Analyzing the data in the literature, we can see that even under the conditions of highly developed healthcare (ex. in Norway), where all three device-aided therapies are available, there are significant differences between regions regarding the use of these therapies [40]. Similarly, significant differences have been reported in a previous Swedish study: the proportion of patients followed by a neurologist ranges from 52% to 96% per province (with an average of 1.7 times per year). 7% of the patients and 5% of APD patients are only treated by the general practitioner (although 38% of APD patients rated their general health as poor or very poor) [41]. Three-quarters of patients had some knowledge of the three device-aided therapy alternatives, but only a quarter indicated that this information was provided by their treating physician.

In light of all this, it is conceivable that in the selection of each device-aided therapies (in centers where all alternatives are available), more emphasis may be placed on the alternative that allows monotherapy (in appropriately selected patients) [36].

At the start of clinical use of LCIG, the recommended treatment duration was a minimum of 16 h [42]. Over the years, starting with higher doses and administration for 24 h was required in more and more patients, in order to achieve the appropriate clinical effect, and this appears to be an appropriate method to significantly improve the patients’ quality of life. Annoying dyskinesias may also be greatly ameliorated by 24-h administration of the LCIG infusion [39,43]. In the present study, the mean duration of administration was 19.8 ± 3.6 h, but in a number of 62 patients we had to administer LCIG for 24 h to maximize the therapeutic benefit. We would like to point out that the “real” doses are in fact those settings (LCIG dose required for 24 h, rate of administration, amount of extra doses, etc.) with which we considered that we obtained the maximum benefit at the time of patient discharge (testing period and adjustment period after PEG-J). New data will probably appear in the future regarding the still unresolved practical issues related to the process of initiating LCIG treatment. A key point from this perspective is the method of calculation of the doses as well as those factors that we will have to take into account during this process (ex. body mass index, the need or subsequent use of the add-on dopaminergic medication, etc.).

Regarding the time of initiation, the titration of LCIG (duration of titration on the nasal-jejunal tube) and the management of previous dopaminergic medication is largely based on the expertise of the treating physician. The introduction of LCIG and trial dosing should be timed in a way that the patient does not feel rushed, but should not be delayed so that the expected efficacy is appropriate. In this way the number of subsequent interruptions can be reduced. One goal would be for the patient to receive the correct dose of LCIG monotherapy because, as COSMOS has demonstrated, this treatment alone can have a sufficiently effective effect and improve compliance. For our patients, the titration time was approximately one week in hospital settings. The COSMOS study (the first study that analyzed co-medication associated with LCIG treatment and investigated the possibility of using LCIG alone) provides evidence that LCIG monotherapy is a feasible and effective long-term option for the treatment of patients with APD [36,44].

A retrospective analysis of patients enrolled in the GLORIA study showed that the adverse event rate was higher in patients requiring doses greater than 2000 mg/day of LCIG, compared with those on doses below 2000 mg/day, although the safety profile and efficacy were acceptable [45]. Regarding the difference between the calculated theoretical dose and the doses with which we finally discharged patients (as a result of a titration process that aimed to maximize the therapeutic benefit) (Figure 1), we consider this as an additional argument in favor of testing the effectiveness of LCIG in conditions of continuous hospitalization.

In our previous article, we followed for 18 months the effectiveness of LCIG in 40 patients with severe disabling dyskinesia and we showed that 4 of the 6 patients in whom treatment was discontinued (including a patient with repeated psychotic episodes) received ≥2000 mg/day LCIG [39]. The tolerance profile of LCIG is considered to be that of orally administered levodopa, except for adverse events related to the delivery system and those given by the presence of gastrostomy [46].

In a recent analysis of the reasons for discontinuation of treatment, we pointed out that when initiating patients on LCIG, it would be useful to identify a “risk profile” for fallout (which is supposed to increase over time). Thus, in patients with multiple comorbidities, in those with cognitive impairment, in those who require significantly higher doses compared to the calculated doses and/or continuous administration for 24 h, in those with dopaminergic side-effects, etc., additional measures such as more frequent evaluations, greater involvement of the aftercare surveillance could extend the long-term therapeutic benefit of LCIG. These observations are consistent with the idea previously held by other experts that the benefits of LCIG treatment might be greater in patients with less advanced disease (in terms of disease duration and severity of motor complications). [47].

## 5. Conclusions

In the present survey, we present the experience of a University Teaching Hospital where the only available device-aided therapy is LCIG. Our previous analyses have demonstrated that Parkinson’s patients in our center are treated according to international guidelines and with a strategy similar to that followed by professionals in other countries [48,49,50]. Due to the fact that treatment options available in Romania are broadly similar to those in other Central- and Eastern-European countries we strongly believe that this type of comprehensive analysis will help both the general neurologist as well as the movement disorder specialist to accurately assess the more advanced stages of Parkinson’s disease and to identify suitable candidate patients for LCIG treatment earlier. We hope that comparing several databases with relevant data from many centers could result in additional information of great practical importance for the increasingly secure use of LCIG. One of the limitations of the study is that the recruitment of hospitalized patients at a University Teaching Hospital, possibly resulted in an increased prevalence of patients with more advanced PD. Another one is the retrospective data processing did not allow for the analysis of some non-motor symptoms (falls, pain, sleep disorders) that significantly affect quality of life, as these were not properly recorded in all cases. Also, the fact that a number of combination therapy options were not available during the evaluated period can be considered an important limitation of the decision-making process of the clinician.

A carefully and individually adjusted LCIG treatment significantly improves the quality of life of APD patients, but questions remain unresolved even after treating a large number of patients for 10 years. We believe that sharing the ideas and observations based on the real-life experience of high-traffic centers can further refine and improve patient selection criteria as well as the practical considerations of the implementation of LCIG treatment (optimal timing, the appropriate dose and duration of administration, and the necessary adjuvant treatment).

## Figures and Tables

**Figure 1 pharmaceutics-14-01115-f001:**
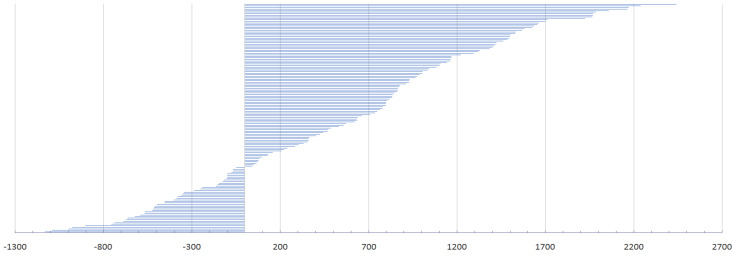
The differences between theoretical and real levodopa/carbidopa intestinal gel doses (mg).

**Figure 2 pharmaceutics-14-01115-f002:**
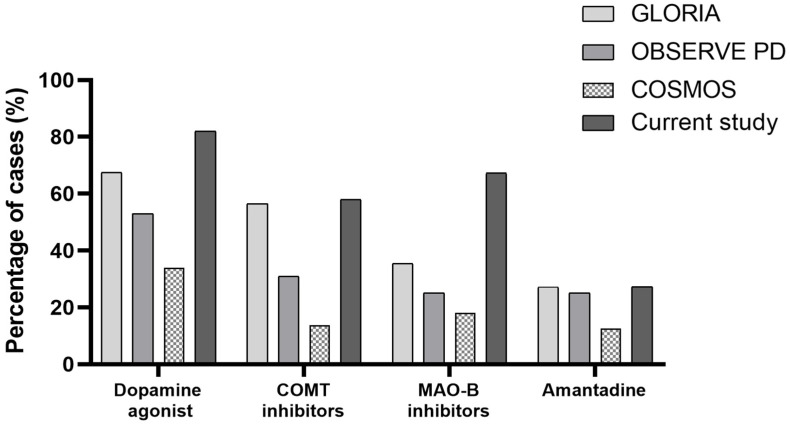
Comparison of add-on therapies in APD patients among some relevant databases and our data.

**Table 1 pharmaceutics-14-01115-t001:** The spectrum of motor complications (fluctuations and different subtypes of dyskinesia).

Motor Complications	n%	Duration (Hours, Mean ± SD)
*off* periods	150 (100%)	4.73 ± 1.08
peak dose dyskinesia	95 (63.3%)	2.96 ± 0.84
diphasic dyskinesia	44 (29.33%)	3 ± 0.82
early morning akinesia	132 (88%)	
delayed *on*	90 (60%)	
no *on*	35 (23.33%)	
sudden *off*	58 (38.67%)	
freezing	81 (54%)	

n = number of patients, SD = standard deviation.

**Table 2 pharmaceutics-14-01115-t002:** Add-on therapies before LCIG initiation.

Add-on Therapy	n%	Dose (mg/Day, Mean ± SD)
Dopamin agonist	123 (82%)	
Pramipexol	42	2.25 ± 0.60
Ropinirol	33	13.45 ± 4.98
Rotigotine	51	8.16 ± 2.93
MAO-Bi (n,%)	101 (67.33%)	
COMTi (n,%)	87 (58%)	
Amantadine (n,%)	41 (27.33%)	

n = number of patients, SD = standard deviation, MAO-Bi = monoamine oxidase B inhibitor, COMTi = catechol-O-methyl transferase inhibitor.

**Table 3 pharmaceutics-14-01115-t003:** Patient characteristics before and after PEG-J.

	Before PEG-J	After PEG-J
	N = 150	N = 150
LD /LCIG dose (mean ± SD)	854.16 ± 258.15 mg	1877 ± 769 mg
Hoehn and Yahr scale (Mean ± SD)		
*on* state	3.23 ± 0.42	2.96 ± 0.2
*off* state	4.39 ± 0.50	3.82 ± 0.4
PGI-I (mean ± SD)	--	1.7 ± 0.55
Very much improved (n)		52
Much improved (n)		91
Minimally improved (n)		7
No change (n)		0
Minimally worse (n)		0

N, n = number of patients, SD = standard deviation, PEG-J = percutaneous endoscopic gastro-jejunostomy, LD = levodopa, LCIG = levodopa/carbidopa intestinal gel, PGI-I = Patient Global Impression of Improvement.

**Table 4 pharmaceutics-14-01115-t004:** The initial parameters and dosing of LCIG therapy.

Characteristics of LCIG Administration	
Titration days (mean ± SD)	6 ± 1 days
LCIG calculated (mean ± SD)	1309 ± 321 mg
LCIG real (mean ± SD)	1877 ± 769 mg
LCIG infusion administration	
Mean ± SD	19.8 ± 3.6 h/day
median	18 h/day
duration of LCIG administration (hrs/day)	n
16 h	53
18 h	35

n = number of patients, SD = standard deviation, LCIG = levodopa/carbidopa intestinal gel, hrs = hours.

## Data Availability

The data presented in this study are available on request from the corresponding author. The data are not publicly available due to privacy reasons.

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
