# Peer review of "Levodopa-Carbidopa Intestinal Gel in Advanced Parkinson’s Disease: Observations and Dilemmas after 10 Years of Real-Life Experience"

_pharmaceutics, 2022, doi:10.3390/pharmaceutics14061115_

Reviewer 4 Report
Although the long-term course of LCIG is very interesting, the method of evaluation of the effects of LCIG in patients is not a scientific evaluation. Evaluation of on-time, off-time, and dyskinesia time is needed. In addition, the authors should indicate whether there is any overlap with their previous reports and forthcoming papers.
Reviewer 5 Report
The Authors present a robust, real world retrospective analysis of LCIG in 150 advanced PD patients. Their findings add value to the established scientific literature on how to contextualize this advanced therapy in the care of Parkinson's patients with motor complications.
The data presented focuses primarily on the state of the cohort and their regimen prior to initiating LCIG. The authors present changes in HY and PGI-I clinical data but do not indicate explicitly at what time point this data was captured after patients were started on LCIG.
How many patients remained on LCIG? What were the reasons for those who discontinued treatment?
What was the breakdown of LCIG treatment duration for the cohort?
Was the "5-2-1" used in selecting this 150 patients for LCIG? The methods do not explain the multidisciplinary approach taken to determine candidates for LCIG? Was DBS an option for this cohort?
讨论范围很广,但不连贯,并且曲折地提出了一系列实践挑战,但没有花足够的时间解释这项研究如何在其中任何一个问题上发挥作用,特别是因为作者提供的队列临床结果数据很少。讨论更像是一本书章节的序言,最好通过解释当前研究的优势和局限性,因为它与概述的实践挑战有关。
作者回应
请参阅附件。
第二轮
审稿人 4 报告
我们指出的区域通常已得到纠正。